

# Manure in combination with optimal topdressing with nitrogen fertiliser improved growth, grain yields and the efficiencies of water and nitrogen use in winter wheat in the Xinjiang Oasis drylands

Yanfei Fang[1], Jianghua Tang[1], Shanqing Zhang[2], Na Zhang[3], Xiaoying Luo[1], Dongping Hu[1] and Wenxiu Xu[1]

[1] College of Agronomy, Xinjiang Agriculture University, Urumqi, Xinjiang, China
[2] Xinjiang Uygur Autonomous Region Agro-meteorological Observatory/Information Centre of Xinjiang Xingnong-Net, Urumqi, China
[3] Institute of Economic Crops, Xinjiang Academy of Agricultural Sciences, Urumqi, China

Corresponding author
Wenxiu Xu, xjxuwenxiu@163.com

## ABSTRACT

**Background**. Currently, low-cost food production using little or no fertiliser is common in oasis dry farming in Xinjiang. This approach results in excessively low crop yields and resource utilisation. Given the limited local precipitation, optimising fertiliser application to improve crop yields, water use efficiency (WUE), and nitrogen use efficiency (NUE) is key. However, the effects of manure and topdressing nitrogen (N) fertiliser on the yields, WUE, and NUE of dryland wheat in the Xinjiang Oasis region of China have not been studied sufficiently.

**Methods**. Therefore, we conducted a 2-year field experiment that examined manure (0 and 30 t ha$^{-1}$ yr$^{-1}$, expressed with M0, M1, respectively) and topdressing urea (0, 150, and 300 kg ha$^{-1}$, expressed with N0, N150, N300, respectively) to quantify the effects of different fertilisation strategies on wheat growth and development, crop N uptake (NUA), soil moisture, yields, WUE, and NUE.

**Results**. Compared with application of chemical N alone, combination with manure increased leaf area index (LAI) and aboveground biomass (ABG) values, and crop NUA. Soil water storage (SWS) increased with soil depth and was 27.5–55.2% higher in the upper soil layer than in the deeper layer. The high evapotranspiration (ET$_c$) caused by adding manure reduced SWS by promoting crop growth, and increased grain yield, WUE, and NUE by 12.9%, 9.8%, and 29.7%, respectively. Compared to no topdressing N treatment, the N150 and N300 treatments significantly increased LAI values (39.8% and 32.8%), ABG (33.0% and 23.7%), NUA (30.4% and 39.4%), ET$_c$ (3.5% and 3.9%), grain yields (16.1% and 10.1%), and WUE (13.7% and 6.8%), while they reduced SWS (8.7% and 9.2%). The interaction effects of manure and nitrogen on LAI, NUA, ET$_c$, and WUE values were significant. The greatest grain yields (2,561.3 and 3,161.2 kg ha$^{-1}$), WUEs (10.8 and 9.5 kg ha$^{-1}$ mm$^{-1}$), and NUEs (32.6% and 43.9%) during the two growing seasons were obtained with the M1N150 treatment. Principal component analysis (PCA) showed that the M1N150 treatment had the highest comprehensive evaluation score.

**Conclusions**. Therefore, we suggest that the combination of 30 t ha$^{-1}$ yr$^{-1}$ manure and 150 kg ha$^{-1}$ topdressing N fertiliser is the optimum fertiliser strategy for improving productivity and efficient water and fertiliser management in dryland winter wheat in the Xinjiang Oasis, where precipitation is low.

# INTRODUCTION

With drylands accounting for 70–75% of the total arable land worldwide and meeting about 60% of the food and nutritional needs of the world's population, the importance of rainfed croplands cannot be over-emphasised (*Biradar et al., 2009*). Nearly one-third of the arable land in China consists of rainfed agricultural areas, and Northwest China is the main rainfed wheat-growing area (*Li, 2004*; *Ren et al., 2022*). Although Xinjiang Province in the northwest is an area of typical oasis-irrigated agriculture, it is still about 6% rainfed cropland, and its production is important for the livelihoods and food security of farming households in the region. However, scarce long-term precipitation (300–500 mm yr$^{-1}$) creates great uncertainty for high stable yields of winter wheat, leading farmers to prioritise low-cost agricultural production; *i.e.,* soil fertility is low because farmers apply little mineral fertiliser and no organic fertilisers, generating a dual water–nitrogen (N) limitation on yields (*Dai et al., 2022*). To compensate for this shortcoming, increased N input has long been recognised as a way to achieve stabilised or increased grain yields rapidly (*Ma et al., 2022*). However, improper use of chemical fertilisers can lead to low fertiliser utilisation and soil environmental problems, posing a threat to sustainable agriculture (*Qiao et al., 2022*; *Zhou et al., 2023*). By contrast, organic fertilisers can significantly increase the soil organic matter content, improve soil structure, promote microbial colonisation, and maintain nutrient balance (*Yang et al., 2025*), and they are far superior to chemical fertilisers for promoting sustainable agricultural practices. Despite these insights, the optimal fertilisation strategy to improve crop yields and efficient resource use in the drought-prone oasis dryland areas of Xinjiang is not clear.

Nitrogen fertiliser application is a versatile method of regulating soil N supply rapidly and significantly increasing crop yields (*Liu et al., 2014*). However, it is often applied in large quantities as a basal fertiliser before sowing, which leads to N shortage during the late growth period of winter wheat due to the N release rate being too rapid, thus reducing the yield-enhancing effects of N and N use efficiency (NUE) (*Yin et al., 2024*; *Zhang et al., 2022*). Topdressing N is a strategic intervention for improving crop yields that provides the N that is essential for subsequent crop growth stages when soil N levels are low (*Ji et al., 2021*). Optimising topdressing N rates benefits crop growth and development, leading to high grain yields and water-N use (*Shen et al., 2024*). It has been proved that the regulation of fertilization rate not only reasonably allows crops to efficiently utilize precipitation and fertilizers in any precipitation year, but also avoids nutrient deficiency or over-application

to crops (*Guo et al., 2012*). Therefore, it is necessary to optimise topdressing N strategies to maximise dryland wheat productivity and water and NUE under the precipitation conditions in that region.

As a traditional source of organic nutrients, manure recycles organic nutrients, improving soil fertility and crop productivity (*Zhou et al., 2022*). Application of manure as fertiliser improves soil porosity, soil aggregates, and physical properties such as soil moisture (*Karami et al., 2011*). Incorporation of organic matter from manure into farmland also increases the abundance and activity of functional microorganisms (*Philippot et al., 2023*), which promote mineralisation of manure (*Cao et al., 2023*). The release of inorganic nutrients ensures a continuous supply of nutrients at all crop growth stages, significantly improving crop growth, yields, and water and nutrient utilisation (*Wang et al., 2019*). However, the positive effects of organic fertiliser application are not absolute; manure has a low mineralisation rate in dryland wheat fields (*Ma et al., 2022*), and the use of organic nutrient management alone would result in incomplete organic availability, and even risk yield reduction (*Chivenge, Vanlauwe & Six, 2011*). Combined application of manure and topdressing N effectively solves this problem.

Addition of organic and chemical fertilisers can optimise soil microbe-driven internal nutrient cycling and regulate soils' physical properties (*Germaine et al., 2010*). Integrating manure with chemical fertilisers has the advantages of efficient long-lasting fertilisation, ensuring sufficient nutrients, while also improving soil water stable macro-aggregates, providing good hydrothermal conditions for wheat growth and increased grain yields and water use efficiency (WUE) (*Rasool, Kukal & Hira, 2008*; *Liu et al., 2013*). Moreover, a meta-analysis of winter wheat in northern China showed that N fertiliser plus organic fertiliser produced better results than partial substitution of chemical fertiliser N by organic fertiliser (*Wang et al., 2020*). Addition of manure and optimal N input significantly improved crop N uptake (NUA), leaf photosynthesis, and aboveground biomass (*Liu et al., 2023*; *Kubar et al., 2022*), leading to a decrease in the total soil water storage (SWS) and nitrate content of dryland cropland used for food production and reducing the negative impact on the environment (*Li et al., 2023*). Similarly, combined application of organic and chemical fertilisers increased NUE and crop productivity *via* organic carbon accumulation (*Pan et al., 2009*). As the agricultural paradigm undergoes transformation, a shift in local fertiliser management towards more sustainable practices is desirable. The combined application of organic and chemical fertilisers, that has become more widely used in agriculture, favours such a shift (*Wang, Tian & Xu, 2023*). Developing a fertiliser management model in low precipitation dry farming areas to improve resource use efficiency is essential for secure crop development, grain yields, and the sustainability of rainfed agriculture.

In summary, although manure and N fertilisers improve crop yields, the results of fertiliser application are affected by climatic conditions, soil properties, crop varieties, and the proportions used in combined application (*Duan et al., 2014*; *Lv et al., 2023*). Furthermore, few studies has focused on the effects of manure and topdressing N rates on yield and water-N efficiencies in dryland wheat, especially under the unique geographic and climatic conditions of the Xinjiang Oasis. This study investigated the effects of manure and

topdressing N rates on the growth, soil water, grain yield, and WUE and NUE of rainfed winter wheat in Xinjiang Oasis, China, and the optimal fertilisation strategy to synergize wheat productivity and the efficient use of water and N.

## MATERIALS & METHODS

### Experimental site description

A field experiment was conducted from September 2021 to July 2023 in Sunjiagou Village, Mulei Kazakh Autonomous County, Changji Prefecture, Xinjiang Uygur Autonomous Region (43°83′N, 90°28′E). The study site is in a semi-arid rainfed agricultural region characterised by typical hilly-mountainous dryland. It is at an altitude of 1,272 m, with an annual average air temperature of 6.6 °C, annual sunshine duration of 3,070 h, frost-free period of 145 days, and average annual rainfall of 354 mm (1990–2023). The rainfall during the 2021–2022 and 2022–2023 growing seasons was 17% and 15% lower than the annual average, respectively. The soil type is dominated by chestnut-calcium (Chinese soil taxonomy). Table 1 details the physical and chemical properties of the 0–20 cm soil layer at the start of the experiment. Figure 1 illustrates the monthly precipitation and average air temperature throughout the experiment. The precipitation at different stages of the experiment is detailed in Table 2.

### Experimental design

The field trial was conducted using a split-plot design with three replicates; each experimental plot measured 60 m$^2$ (6 m × 10 m). Manure was applied to the main plots at either 0 (M0) or 30 t ha$^{-1}$ yr$^{-1}$ (M1). The subplots consisted of three nitrogen application rates: 0 (N0), 150 (N150), and 300 (N300) kg ha$^{-1}$ urea (urea, 46% N). The manure used was cow dung decomposed by local herdsmen, with average nutrient contents of: total nitrogen, 18.02 g kg$^{-1}$; total phosphorus, 21.15 g kg$^{-1}$; total potassium, 36.35 g kg$^{-1}$; and organic matter, 360.40 g kg$^{-1}$. Both the cattle manure and 150 kg P$_2$O$_5$ (44% P) were applied as a basal fertiliser before ploughing, and were incorporated into the soil through tilling. Topdressing nitrogen fertilizer for dryland winter wheat was applied twice in season, once coinciding with early spring snow-melt (on March 26, 2022 (stage 16 of Zadoks' scale, (*Zadoks, Chang & Konzak, 1974*)) and May 31, 2023 (stage 58 of Zadoks' scale)) and in combination with rainfall during the jointing to anthesis stages (on April 9, 2022 (stage 18 of Zadoks' scale) and May 22, 2023 (stage 47 of Zadoks' scale)). The wheat variety used in this experiment was Xingdong 18, sown at a depth of five cm, with a row spacing of 15 cm, and a seeding rate of 225 kg ha$^{-1}$. After harvesting in mid to late July, the plots were tilled in early August following rainfall, and rotary tillage and land levelling were performed before planting. There was no tillage between sowing and maturity. The wheat was sown on September 20, 2021, and October 4, 2022, and harvested on July 6, 2022, and July 15, 2023, respectively. The preceding crop at the start of the 2-year experiment was chickpea. Irrigation was not used in any experimental years. Weed, pests and diseases were controlled by spraying herbicides and pesticides during the regreen stage, consistent with local farming practices.
**Table 1** Properties of the 0–20 cm soil layer sampled in the experimental field at the sowing of winter wheat in 2021–2022 and 2022–2023.

| Years | SWC (%) | pH (H$_2$O) | SOM (g kg$^{-1}$) | STN (g kg$^{-1}$) | SAP (mg kg$^{-1}$) | SAK (mg kg$^{-1}$) |
|---|---|---|---|---|---|---|
| 2021–2022 | 10.9 | 8.2 | 14.9 | 1.2 | 13.1 | 283.3 |
| 2022–2023 | 14.0 | 8.1 | 15.6 | 1.1 | 14.3 | 305.7 |

Notes.

SWC, soil water content; the pH of water was determined with a 1:2.5 soil to water ratio; SOM, soil organic matter (dichromate wet oxidation); STN, soil total nitrogen (Kjeldahl method); SAP, soil available phosphorus (molybdate-ascorbic acid method); SAK, soil available potassium (extraction with 1 mol L$^{-1}$ NH$_4$ OAc and analysed by flame photometry). The methods follow *Bao (2000)*.

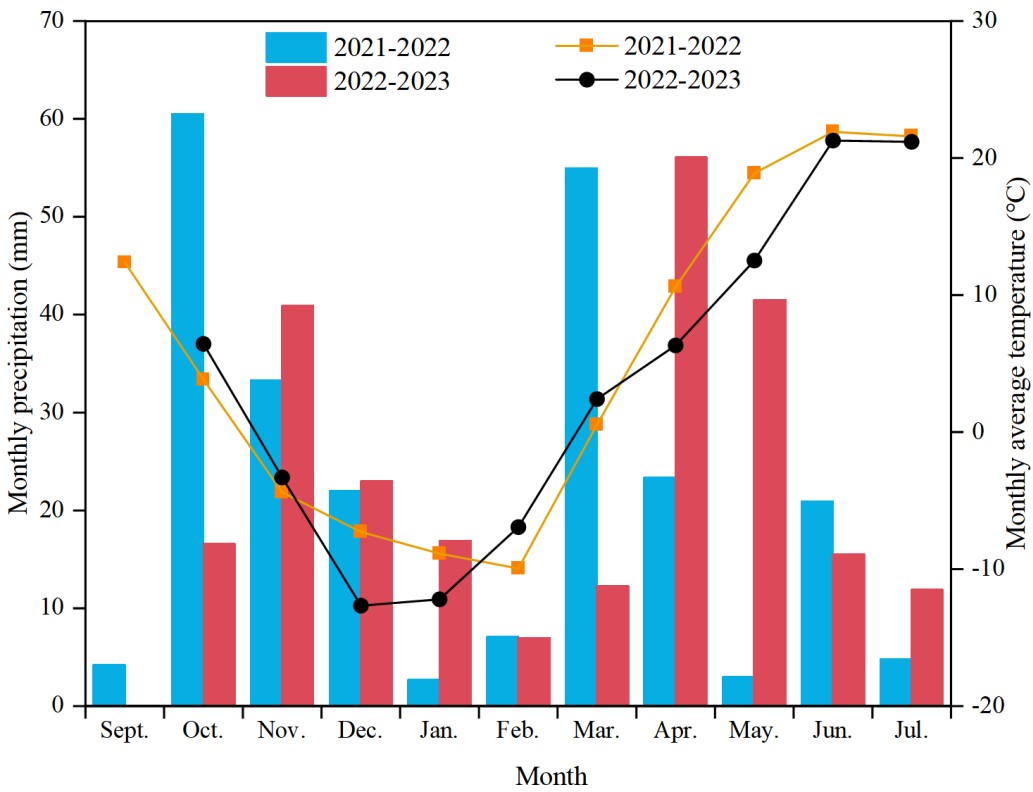

**Figure 1** Monthly rainfall and average air temperature of dryland wheat during the 2021–2023 growing seasons in Mulei Kazakh Autonomous County, Changji Prefecture, Xinjiang, China.

**Table 2** Precipitation at different stages at the study site (mm).

| Year | FS | SS-JS | JS-AS | AS-MS | Total |
|---|---|---|---|---|---|
| 2021–2022 | 56.8 | 210.7 | 0.5 | 25.7 | 293.7 |
| 2022–2023 | 59.3 | 190.4 | 23.9 | 27.4 | 301.0 |

Notes.

FS, Precipitation during the fallow stage; SS-JS, sowing stage to jointing stage; JS-AS, jointing stage to anthesis stage; AS-MS, anthesis stage to maturity; Total, total precipitation.

## Sampling and measurement
### Grain yield and yield components
At the winter wheat mature stage (stage 94 of Zadoks' scale), three representative quadrats covering 1 m² (1 m × 1 m) were randomly harvested from each plot and air-dried for grain separation. Grain from the three sample squares in the same plot were mixed and weighed to determine grain yield. At physiological maturity (stage 92 of Zadoks' scale), spike number was counted from the two one m double rows of each plot. Grain number per spike and 1,000-grain weight were determined by counting the grains of 50 randomly selected spikes from the two, one m double rows.

### Soil water content and storage
Soil cores four cm in diameter were randomly collected from each soil layer (each layer was 20 cm thick, for a total of 100 cm) at three locations in each plot at the pre-sowing, anthesis (stage 64 of Zadoks' scale), and maturity (stage 94 of Zadoks' scale) stages in each growing season using a hand-held soil auger, and the values were averaged. The soil samples were packed in aluminum boxes, brought to the laboratory immediately, and dried in an oven at 105 °C for 24 h to a constant weight, to determine the soil moisture content (soil water content (SWC), %). Undisturbed soil cores (100 cubic centimeters) were collected from soil profile pits at depths of 0–100 cm to determine the soil bulk density following *Finn et al. (2015)*. Soil water storage (SWS) was calculated as follows (*Mo et al., 2016*):

$$SWS\ (mm) = SWC\ (\%) \times \rho b\ (\text{g cm}^{-3}) \times SD\ (mm) \tag{1}$$

where $\rho b$ (g cm$^{-3}$) is the dry soil bulk density of a given soil layer, and $SD$ (mm) is the given soil depth.

## Evapotranspiration and water use efficiency
The actual evapotranspiration ($ET_c$, mm) of winter wheat was estimated using the soil water balance equation. $ET_c$ was calculated using the following formula (*Li et al., 2023*):

$$ET_c = P + I + U + (SWS_2 - SWS_1) - R - D \tag{2}$$

where $SWS_1$ (mm) is the soil water storage at the time of sowing, $SWS_2$ (mm) is the soil water storage at the end of the growing season, and $P$ is the total rainfall during the wheat growing season. $I$, $R$, $U$, and $D$ are the amounts of irrigation (mm), surface runoff (mm), upward capillary flow into the root zone (mm), and drainage (mm), respectively. As the experimental site is rainfed farmland with a deep groundwater table and no irrigation was provided over the growing season, $I$, $R$, $U$, and $D$ are negligible.

Finally, water use efficiency (WUE) was calculated using the following formula (*Zhao et al., 2023*):

$$WUE\ (\text{kg ha}^{-1}\ \text{mm}^{-1}) = Y/ET_c \tag{3}$$

where $Y$ (kg ha$^{-1}$) is the grain yield and $ET_c$ (mm) is the actual evapotranspiration during the crop growing season.

## Leaf area index and photosynthetic parameters

At the jointing (stage 31 of Zadoks' scale), booting (stage 43 of Zadoks' scale), anthesis (stage 64 of Zadoks' scale), early milk (stage 73 of Zadoks' scale) and early dough (stage 83 of Zadoks' scale) stages, the leaf area index (LAI) was measured from 10 randomly selected destructive plant samples. Leaf length (L) and the widest width (W) of each leaf of all 10 sampled plants were measured with a ruler, and the LAI was calculated as follows (*Gao et al., 2010*):

$$LAI = 0.8 \times \frac{\sum_{i=1}^{m} \sum_{j=1}^{n} (Lij \times Wij)}{m} \times N/S \tag{4}$$

where, $Lij$ is the leaf length (cm) of the $jth$ leaf on ith plant, $Wij$ is the largest width (cm) of the $jth$ leaf on ith plant, $m$ ($m = 10$) is the measured number of plants, $n$ is the number of leaves per plant, $N$ is the plant numbers of a plots, S is land area of a plot ($cm^2$).

The fag leaf from the main tillers of five random plants was tagged at anthesis. The net photosynthetic rate ($Pn$) and transpiration rate ($Tr$) of flag leaves of winter wheat at the anthesis stage was measured by the TARGAS-1 Portable Photosynthesis System (PP Systems) between 9:00 and 11:00 on sunny days. The $CO_2$ concentration of the leaf chamber was set as 400 $\mu mol\ mol^{-1}$ and the other gas exchange parameters (1,000 $\mu mol \cdot m^{-2} \cdot s^{-1}$, 25 °C). The average value was computed from three flag leaves for each replicate.

## Aboveground biomass, crop N uptake and nitrogen use efficiency

Fifteen plants were sampled from each plot at the maturity (stage 94 of Zadoks' scale) stage. The root system was cut off and the plants were processed further. Plants at the maturity stage were separated into stem + sheath, leaf, glumes, and grains. The plant samples were dried at 105 °C for 30 min, and then at 80 °C until they reached a constant weight. Then, the samples were ground so that they could pass through an 80-mesh sieve and digested using $H_2SO_4 \cdot H_2O_2$. The nitrogen content was determined using Nessler's colorimetric method (*Bao, 2000*). The N uptake (kg $ha^{-1}$) by wheat was quantified by multiplying the biomass of each organ by its respective N concentration, calculated as follows (*Li et al., 2021*):

$$Crop\ N\ uptake = \sum N_c \times O_d \times 10^{-3} \tag{5}$$

where $N_c$ and $O_d$ are the N concentration (g $kg^{-1}$) and dry weight (kg $ha^{-1}$) in different organs of the wheat plant, and $10^{-3}$ is the conversion coefficient. Nitrogen use efficiency (NUE, %) was determined using the recovery efficiency of nitrogen, calculated as follows *Liu et al. (2023)*:

$$NUE\ (\%) = (TA_N - TA_0)/N \tag{6}$$

where $TA_N$ is the total aboveground nitrogen accumulation by wheat with nitrogen fertiliser applied; $TA_0$ is the total nitrogen accumulation by wheat without nitrogen fertiliser applied, and $N$ is the amount of nitrogen fertiliser applied.

## Statistical analysis

Analysis of variance (ANOVA) and principal component analysis (PCA) were performed using IBM SPSS 19.0 (SPSS, Chicago, IL, USA). Correlation analysis and figures were

created with the Origin 2021 software (Origin Lab, USA). Duncan's new multiple-range test was used to test differences at the 0.05 significance level. PCA mapping was also carried out using the Origin 2021 software.

# RESULTS

## Crop growth and development

### Leaf area index (LAI) and aboveground biomass

Manure, topdressing N fertiliser, and year had significant effects on the wheat LAI and aboveground biomass (Table S1). Adding manure fertiliser increased LAI values by 33.5% and aboveground biomass by 25.9% (2-year averages) compared with no manure fertiliser (Fig. 2). With an increased topdressing N rate but without addition of manure, LAI values increased by 35.9% in 2021–22 and aboveground biomass increased by 29.6% (Fig. 2). In 2022–23, the LAI and aboveground biomass values were highest with the M0N150 treatment, significantly increasing by 66.90% and 27.8%, respectively ($P < 0.05$), compared to the M0N0 treatment and by 7.00% and 5.8%, respectively, compared to the M0N300 treatment ($P > 0.05$, Fig. 2). Under manure addition, the highest LAI and aboveground biomass values were with M1N150 in both years (Fig. 2). Compared with the M1N150 treatment, the LAI values of the M1N0 and M1N300 treatments decreased by 35.5% and 9.2% (2-year average, Fig. 2A), respectively, and the aboveground biomass values decreased by 33.3% and 9.9%, (2-year average, Fig. 2B), respectively.

### Photosynthetic parameters

Nitrogen application had a significant effect on the net photosynthetic rate ($Pn$); manure and N had significant effects on the transpiration rate ($Tr$); and year had a significant effect on both $Pn$ and $Tr$. However, there was no significant interaction among the three factors (Table S1). Addition of manure reduced $Pn$ values by 6.8% and $Tr$ values by 3.3% in 2021–22 compared to without manure (Fig. 3). Notably, in 2022–23, addition of manure increased $Pn$ values by 6.74% and $Tr$ values by 17.13% compared to without manure (Fig. 3). Regardless of whether manure was added, the N150 treatment resulted in higher $Pn$ and $Tr$ values compared to the N0 and N300 treatments in both years (Fig. 3). Comparing the $Pn$ and $Tr$ values of each treatment, those of the M1N150 treatment had the highest values. The M1N150 treatment increased the respective $Pn$ values in the M0N0, M0N150, M0N300, M1N0, and M1N300 treatments by 7.5%, 0.1%, 4.4%, 14.3%, 22.6% (2021–22), and by 28.8%, 11.8%, 12.5%, 20.9%, and 10.5% (2022–23); and increased the respective $Tr$ values by 6.9%, 3.3%, 5.7%, 9.3%, 18.8% (2021–22), and by 35.1%, 16.7%, 22.1%, 1.8%, and 18.0% (2022–23) (Fig. 3).

## Grain yield, its components, and NUE

### Grain yield and components

Manure, N, and year significantly influenced wheat grain yields, with no observed significant interaction of these factors (Table 3). Adding manure in combination with fertiliser increased grain yields by 16.7% in 2021–22 and by 9.2% in 2022–23 compared with adding fertiliser alone (Table 3). Regardless of whether manure was added, the N150 treatment at

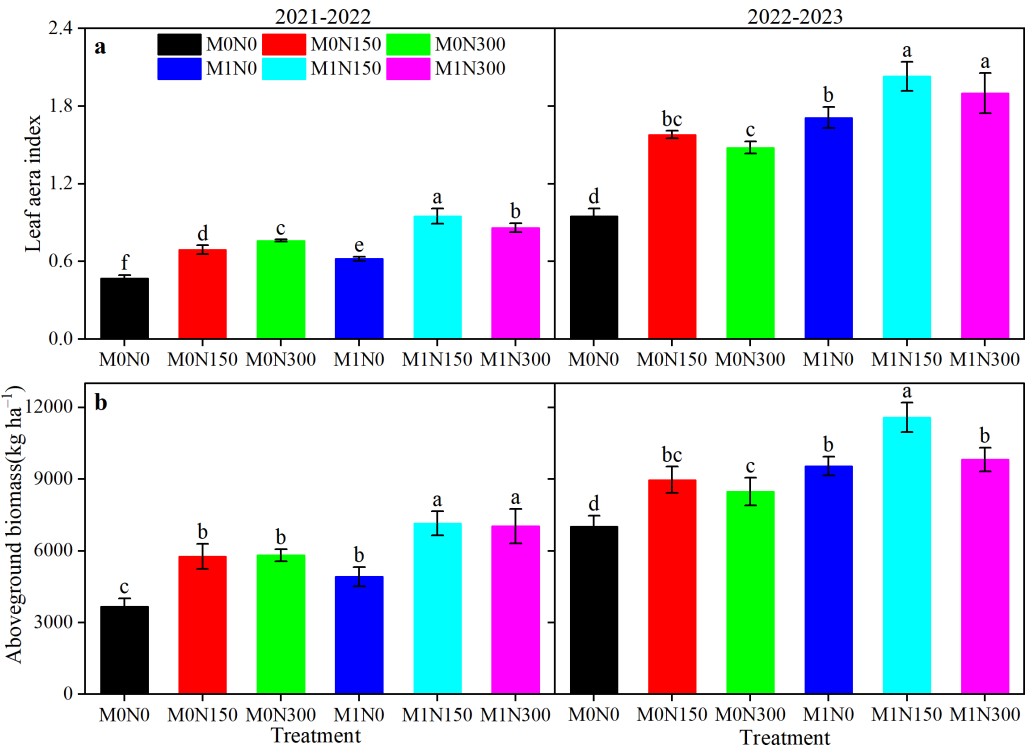

**Figure 2** **Effect of added manure fertiliser and nitrogen application on LAI (A) and aboveground biomass (B) in dryland wheat.** No manure (M0) was added and 0 (N0), 150 (N150), and 300 kg ha$^{-1}$ (N300) of N fertiliser was applied. The addition of manure (M1) and topdressing application of 0 (N0), 150 (N150) and 300 kg ha$^{-1}$ (N300) of nitrogen fertiliser. LAI, mean leaf area index at jointing- early dough stage. The error bar represents the standard deviation of each index in each treatment. Different lowercase letters represent significant differences ($P < 0.05$) between the different fertiliser treatments.

different topdressing N rates increased grain yields over the 2 years by an average of 16.1% compared to N0, and by 5.6% compared to N300 (Table 3). Comparing the grain yields of each treatment, the highest was recorded with the M1N150 treatment, with yields of 2,516.3 kg ha$^{-1}$ in 2021–22 and 3,161.4 kg ha$^{-1}$ in 2022–23; these values were significantly higher than with the other treatments in both years, by 10.4–55.5% in 2021–22 and by 4.2–20.3% in 2022–23 (Table 3).

Manure, N, and year all had significant effects on yields (Table 3). Notably, the interaction between manure and N had very significant effects ($P < 0.01$) on the spike numbers (SNs) and thousand grain weights (TGWs) of wheat, and the three-way interaction had significant effects ($P < 0.05$) on the grain numbers per spike (GNSs) and TGWs (Table 3). Addition of manure increased SN, GNS, and TGW values by averages of 17.7% (2–year average), 4.5% (2–year average), and 0.8% (2021–22), respectively, compared to scenarios without manure, albeit with a corresponding decrease in TGW values by 8.9% in 2022–23. Under different N topdressing rates, the order of SNs (2 years) and GNSs (2022–23) of winter wheat was N0 < N300 < N150 (Table 3). The SNs (2 years) and GNSs (2022–23) of each treatment were compared; the highest values were with M1N150, with average increases

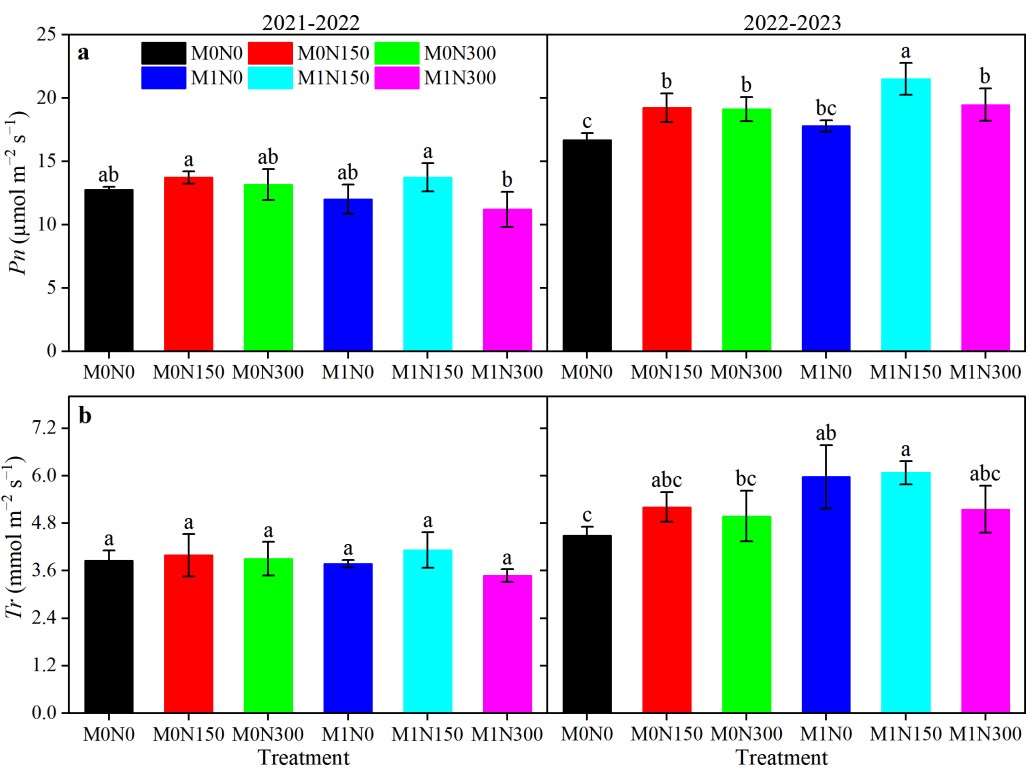

**Figure 3** **Effect of added manure fertiliser and nitrogen application on *Pn* (A) and *Tr* (B) in dryland wheat.** *Pn*, net photosynthetic rate; *Tr*, transpiration rate. The error bar represents the standard deviation of each index in each treatment. Different lowercase letters represent significant differences ($P < 0.05$) between the different fertiliser treatments.

of 5.9–47.5% and 2.0–17.9%, respectively, relative to the other treatments (Table 3). With topdressing N alone, there was no significant difference among the SN rates. But TGW values were significantly reduced in both years and GNS values were significantly reduced in 2021–22, with the N150 treatment resulting in the lowest values. TGWs were highest with the M0N0 treatment in both years, with significant increases of 8.7–12.2% on average over the other treatments (Table 3). In addition, the GNS values of the M1N0 treatment were the highest in 2021–22, with average increases of 2.5–11.3% compared to the other treatments (Table 3).

### Crop NUA and NUE

Manure, N topdressing, and year had very significant ($P < 0.01$) effects on crop NUA and NUE (Table S1). The interaction between manure and topdressing N was very significant only for NUE ($P < 0.01$) (Table S1). Compared to N fertiliser alone, manure fertiliser significantly increased NUA by averages of 25.1% in 2021–22 and 34.4% in 2022–23 (Fig. 4A). N application increased NUA both with and without added manure (M0 and M1 conditions), with greater N application resulting in higher NUA. Both the N300 and N150 treatments differed significantly from the N0 treatment. The highest NUA value was for M1N300, which increased significantly by averages of 38.2% in 2021–22 and 31.3%

**Table 3  Grain yield and yield components under different fertiliser treatments during the 2021–2022 and 2022–2023 growing seasons.**

| Years | Treatment | Spike numbers ($\times 10^4$ ha$^{-1}$) | Grain number (spike$^{-1}$) | TGW (g) | Grain yield (kg ha$^{-1}$) |
|---|---|---|---|---|---|
| | M0N0 | 136.9 ± 12.0 d | 39.3 ± 0.5 a | 31.1 ± 0.8 a | 1,618.3 ± 111.6 c |
| | M0N150 | 204.5 ± 14.7 bc | 35.7 ± 0.8 c | 28.4 ± 0.4 b | 2,220.2 ± 125.3 b |
| 2021–2022 | M0N300 | 198.7 ± 6.1 bc | 36.1 ± 0.9 c | 28.2 ± 0.7 b | 2,083.5 ± 109.0 b |
| | M1N0 | 183.6 ± 12.1 c | 40.3 ± 0.5 a | 30.6 ± 0.7 a | 2,113.7 ± 92.3 b |
| | M1N150 | 232.5 ± 8.7 a | 37.6 ± 0.9 b | 28.5 ± 0.5 b | 2,516.3 ± 97.5 a |
| | M1N300 | 213.3 ± 12.2 ab | 39.0 ± 1.2 ab | 29.2 ± 0.4 b | 2,278.7 ± 95.3 b |
| | M0N0 | 339.2 ± 5.9 c | 30.7 ± 1.1 d | 27.3 ± 0.7 a | 2,628.5 ± 113.0 c |
| | M0N150 | 412.5 ± 17.3 b | 35.6 ± 0.4 ab | 23.5 ± 0.6 c | 2,902.1 ± 108.9 b |
| 2022–2023 | M0N300 | 403.2 ± 12.0 b | 34.0 ± 1.3 bc | 25.0 ± 0.4 b | 2,836.3 ± 115.8 b |
| | M1N0 | 454.4 ± 23.1 a | 33.3 ± 0.9 c | 22.7 ± 0.4 c | 2,941.3 ± 93.7 b |
| | M1N150 | 469.7 ± 14.3 a | 36.3 ± 0.9 a | 22.7 ± 0.6 c | 3,161.2 ± 102.9 a |
| | M1N300 | 449.4 ± 16.3 a | 34.6 ± 1.1 abc | 23.6 ± 0.5 c | 3,033.6 ± 84.9 ab |
| | | *F*-value | | | |
| M | | 126.10[**] | 27.40[**] | 27.87[**] | 70.22[**] |
| N | | 44.96[**] | 0.64 ns | 42.90[**] | 39.07[**] |
| Y | | 2,455.16[**] | 163.03[**] | 729.26[**] | 497.11[**] |
| M × N | | 11.07[**] | 0.28ns | 15.77[**] | 3.00ns |
| M × N × Y | | 1.94ns | 3.52[*] | 5.22[*] | 0.65ns |

Notes.
Different small letters in the same column within a year mean significant differences among treatments at $P < 0.05$.
ns, no significant difference.
[*]$P < 0.05$.
[**]$P < 0.01$.

in 2022–23 over the other treatments (Fig. 4A). Addition of manure promoted NUA with a corresponding increase in NUE (35.1% higher in 2021–22 and 19.8% higher in 2022–23) compared to no manure (Fig. 4). However, increased N fertiliser application led to reductions in NUE (26.3% lower in 2021–22 and 39.8% lower in 2022–23) (Fig. 4B). The highest NUE values of 31.9% and 46.5% were recorded with M1N150 in the 2021–22 and 2022–23 seasons, respectively (Fig. 4B). The lowest NUE values were with M0N300 and were 17.6% in 2021–22 and 23.9% in 2022–23 (Fig. 4B). Specifically, the M1N150 treatment resulted in significantly different values from those of the other treatments over the 2 years.

## Distribution of SWS

In both years, SWS increased with the deepening of the soil layer at anthesis and maturity, and SWS was lower in the upper soil layer (0–40 cm) than in the deep soil layer (40–100 cm). Averaged over all manure and N application rate treatments, the SWS values were 52.5%, 55.2%, 39.6%, and 27.5% lower in the upper soil layer (0–40 cm) than in the deep soil layer (40–100 cm) at anthesis in 2021–22 (Fig. 5A), maturity in 2021–22 (Fig. 5B), anthesis in 2022–23 (Fig. 5C), and maturity in 2022–23 (Fig. 5D), respectively. SWS values were lower in the manure-added treatment than in the non-manure-added

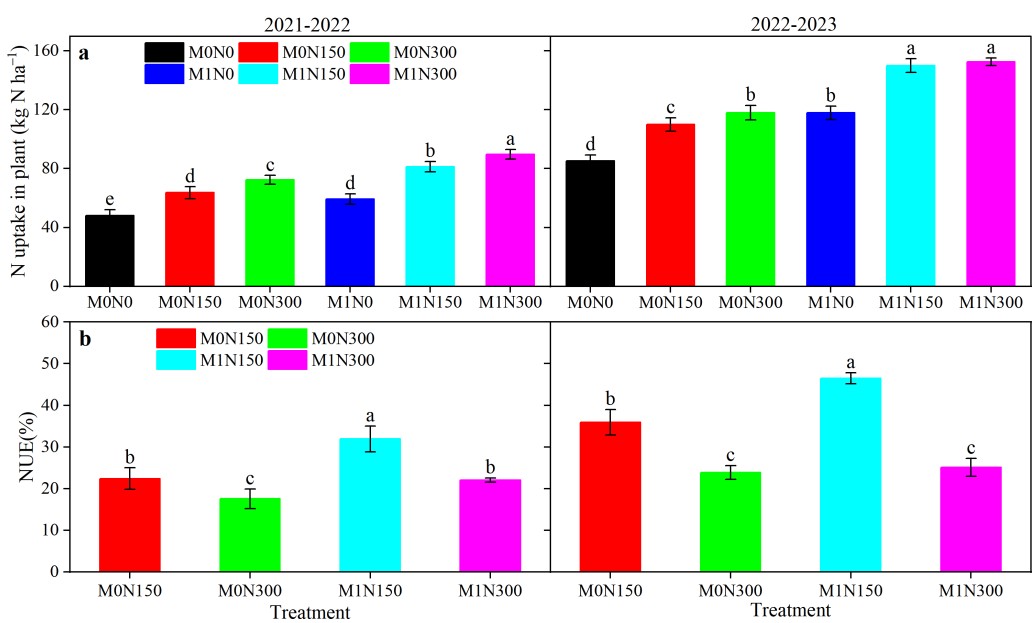

**Figure 4** **Effect of added manure fertiliser and nitrogen application on N uptake in plant (a) and NUE (b) in dryland wheat.** NUE, nitrogen use efficiency. Different lowercase letters represent significant differences ($P < 0.05$) between the different fertiliser treatments.

treatment. The average SWS values were lower in the manure-added treatment than in the non-manure-added treatment by 8.0% in 2021–22 and by 6.9% in 2022–23. Regardless of whether manure was added, the topdressing N fertiliser treatment reduced SWS in the 0–100 cm soil layer compared to no N treatment. Compared to the N0 treatment, average SWS values were 11.0% and 5.4% higher in the N150 treatment in 2021–22 and 2022–23, respectively, and 11.6% and 5.9% higher in the N300 treatment in 22021–22 and 2022–23, respectively. Among all treatments, M0N0 resulted in the highest SWS values and M1N300 resulted in the lowest; the SWS values of the M1N300 treatment were 6.1–22.6% lower than the other treatments at anthesis in 2021–22 (Fig. 5A), 7.3–20.7% lower at maturity in 2021–22 (Fig. 5B), 3.1–14.8% lower at anthesis in 2022–23 (Fig. 5C), and 6.7–18.1% lower at maturity in 2022–23 (Fig. 5D).

## Evapotranspiration ($ET_c$) and WUE

Evapotranspiration ($ET_c$) and WUE were strongly influenced by manure, topdressing N rate, and year (Table 4). The interaction of manure and N fertiliser had significant ($P < 0.05$) and very significant ($P < 0.01$) effects on $ET_c$ and WUE, respectively (Table 4). Addition of manure resulted in average increases in $ET_c$ of 2.83% and in WUE of 9.80% compared to scenarios without manure addition (Table 4). Without manure addition, the overall performance of $ET_c$ with increased N was N150 > N300 > N0. However, with manure addition, $ET_c$ continued to increase with the N rate, to a maximum value in the M1N300 treatment, which increased $ET_c$ values relative to the other treatments by an average of 6.5% in 2021–22 and 3.7% in 2022–23 (Table 4). While the M1N300 treatment results differed significantly from those of the other treatments ($P < 0.05$), the results of the

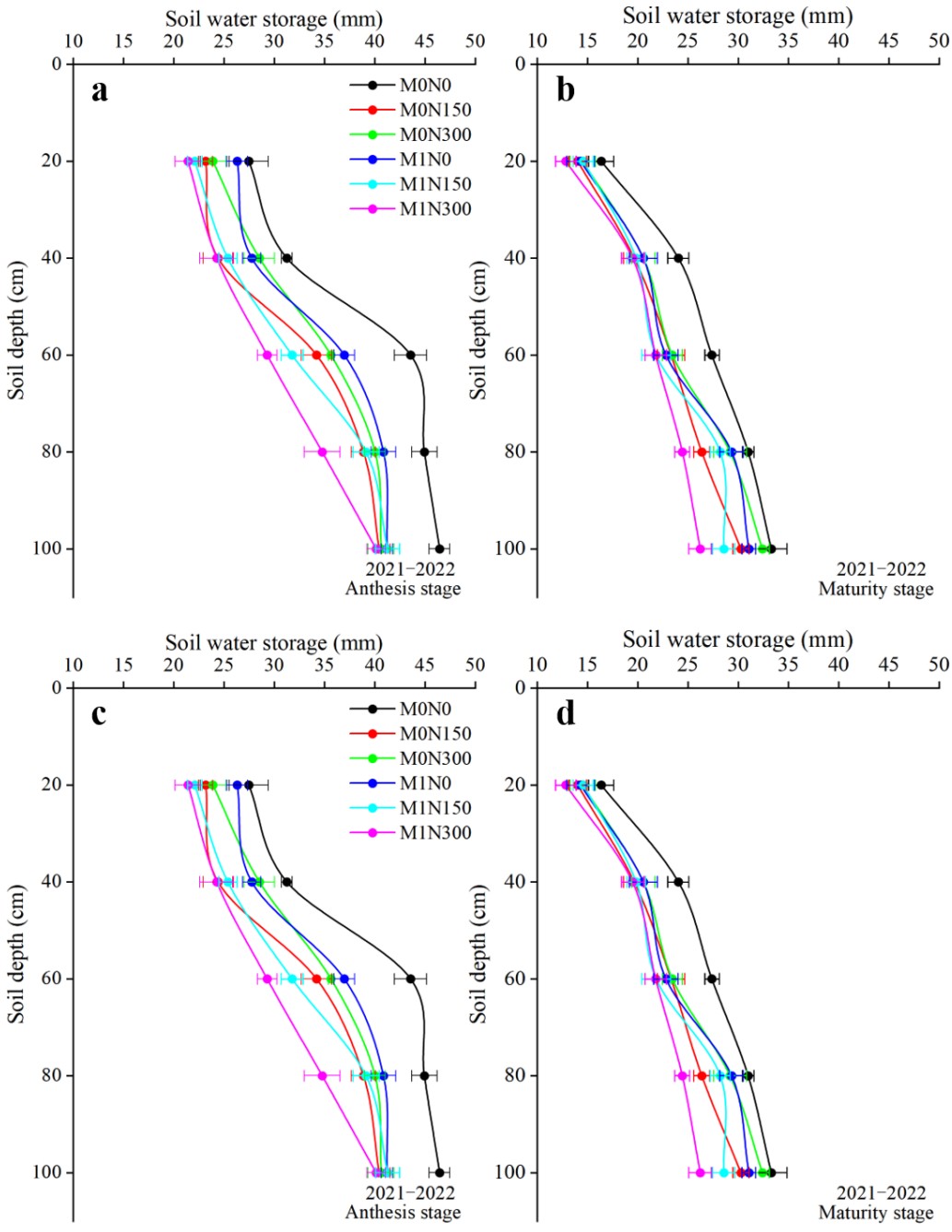

**Figure 5** **Vertical distribution of soil water storage (SWS) in 0–100 cm soil layer at anthesis and maturity stages in the 2021–2022 and 2022–2023 growing seasons.** The horizontal bars indicate standard deviations.

M0N150, M0N300, M1N0, and M1N150 treatments did not differ significantly between the 2 years. Nevertheless, the N150 treatment resulted in higher WUE values compared to the N0 and N300 treatments, with or without the addition of manure. In addition, in 2021–22 and 2022–23, the highest WUE values of 10.8% and 9.5%, respectively, were

**Table 4** ET$_c$ and WUE under different fertiliser treatments during the 2021–2022 and 2022–2023 growing seasons.

| Treatment | ET$_c$ (mm) | | WUE (kg ha$^{-1}$ mm$^{-1}$) | |
|---|---|---|---|---|
| | 2021–2022 | 2022–2023 | 2021–2022 | 2022–2023 |
| M0N0 | 213.39 ± 7.16 c | 324.90 ± 7.84 c | 7.58 ± 0.35 c | 8.10 ± 0.51 b |
| M0N150 | 231.83 ± 9.53 b | 337.91 ± 7.72 ab | 9.57 ± 0.21 b | 8.60 ± 0.49 b |
| M0N300 | 225.33 ± 6.72 b | 330.39 ± 8.07 bc | 9.24 ± 0.27 b | 8.59 ± 0.34 b |
| M1N0 | 227.38 ± 8.52 b | 332.25 ± 7.91 bc | 9.29 ± 0.07 b | 8.86 ± 0.44 ab |
| M1N150 | 232.72 ± 8.93 b | 333.59 ± 7.71 bc | 10.81 ± 0.05 a | 9.48 ± 0.51 a |
| M1N300 | 240.67 ± 9.63 a | 344.17 ± 6.33 a | 9.48 ± 0.51 b | 8.81 ± 0.11 ab |
| | *F*-value | | | |
| M | 8.50** | | 48.72** | |
| N | 6.33** | | 30.61** | |
| Y | 1,534.51** | | 23.90** | |
| M × N | 3.33* | | 6.62** | |
| M × N × Y | 0.08ns | | 1.32ns | |

**Notes.**
ET$_c$, the actual evapotranspiration; WUE, water use efficiency.
Different small letters in the same column mean significant differences among treatments at $P < 0.05$. ns, no significant difference.
*$P < 0.05$.
**$P < 0.01$.

recorded for the M1N150 treatment, and averaged 20.6% and 10.5% higher, respectively, compared to the other treatments (Table 4).

## Relationships between crop growth indicators, crop NUA, soil moisture and yields, yield components, and water and NUE

PC1 and PC2 explained 62.3% and 17.7% of the variance in 2021–22, respectively, and 66.1% and 12.6% of the variance in 2022–23, respectively (Fig. 6A). In the PCA plots (Fig. 6A), grain yield (GY), NUE, and WUE clustered with SN, NUA, LAI, and aboveground biomass (ABG) in both years, with significant ($P < 0.05$) or very significant ($P < 0.01$) positive correlations between the various indicators (Fig. 6B). The positive correlations of LAI and ABG values with GY, NUE, and WUE values were even higher, with correlation coefficients of 0.83–0.87 and 0.61–0.91 for the respective years (Fig. 6B). Significant ($P < 0.05$) or very significant ($P < 0.01$) negative correlations were found between GY, SN, NUE, and WUE values and TGW and ∆SWS values in 2021–22 and 2022–23 (Fig. 6B). In addition, the 2-year performance differed, in that ET$_c$ values were more significantly positively correlated with GY, SN, NUE, WUE, NUA, LAI, and ABG values in 2021–22, while *Pn* and *Tr* values were more significantly positively correlated with these indicators in 2022–23 (Fig. 6B). GNS results were the opposite in both years, with negative correlations with GY, SN, NUE, NUA, LAI, ABG, WUE, ET$_c$, *Pn*, and *Tr* values in 2021–22, and significant ($P < 0.05$) or very significant ($P < 0.01$) positive correlations with these factors in 2022–23 (Fig. 6B). Furthermore, combining the principal component scores showed that the M1N150 treatment resulted in the highest scores in both 2021–22 and 2022–23 (Table S2).

 

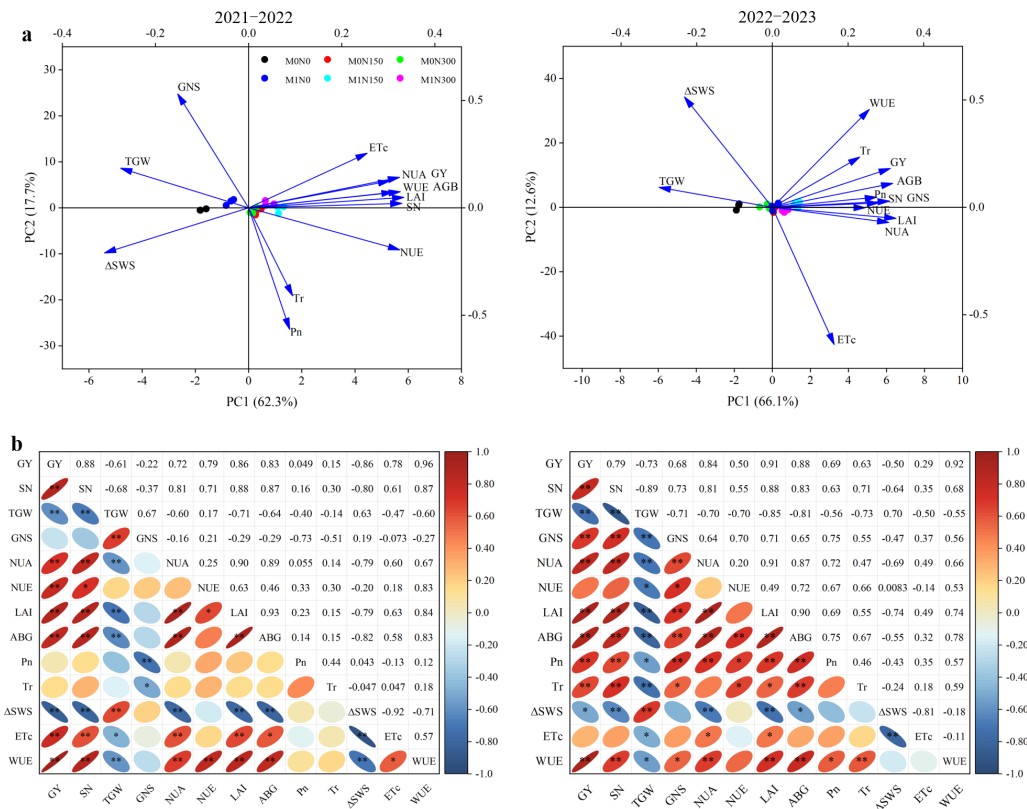

**Figure 6** Principal component analysis (PCA) (A) and correlation analysis (B) of all indicators in in the 2021–2022 and 2022–2023 growing seasons. GY, grain yield; SN, spike number; TGW, thousand grain weight; GNS, grain number per spike; NUA, nitrogen uptake in plant ; NUE, nitrogen use efficiency; LAI, leaf area index; ABG, aboveground biomass; Pn, net photosynthetic rate; Tr, transpiration rate; $\Delta$SWS, mean soil water storage at anthesis and maturity stages; $ET_c$, the actual evapotranspiration; WUE, water use efficiency; **, Correlation is significant at the 0.01 level; *, Correlation is significant at the 0.05 level.

# DISCUSSION

## Impact of fertiliser application on LAI, aboveground biomass, and photosynthetic parameters

LAI and aboveground biomass values are important indicators of crop growth. Previous studies found that LAI values and biomass production were positively affected by manure and N availability (*Kubar et al., 2022*), generally supporting our results. In our study, manure application markedly improved the average LAI value by 33.5% and aboveground biomass by 25.9% (Fig. 2). Furthermore, combination of manure and topdressing N fertiliser augmented LAI values and aboveground biomass (Fig. 2). These results may have arisen because combined application of manure and N fertiliser has the advantages of both organic fertiliser persistence and N fertiliser availability, providing both sufficient nutrients and good water, air, and heat conditions for crop growth (*Li et al., 2025*), all of which improve the single leaf area and population LAI values of wheat. A greater leaf area increases light interception and ultimately produces more biomass (*Si et al., 2020*).

However, N application has a threshold effect in regulating crop growth (*Wang et al., 2017*); excessive N input may be associated with root growth inhibition and difficulty with nutrient uptake and utilisation, which are ultimately detrimental to crop growth and development (*Liu et al., 2017*; *Si et al., 2020*). This generally supports our finding that the M1N300 treatment reduced LAI and aboveground biomass values compared to the M1N150 treatment (Fig. 2).

Our study revealed that the use of manure increased *Pn* and *Tr* values in the second year compared to no manure (Fig. 3). This aligns with previous reports that organic fertilisers directly supported chlorophyll synthesis by enhancing the availability of these nutrients, thereby improving the overall photosynthetic capacity of the plant (*Mthiyane, Aycan & Mitsui, 2024*). Notably, overall, the manure treatments reduced *Pn* and *Tr* values in the first year, with the M1N300 treatment resulting in the lowest values. We postulate that this was related to drought stress in the first year (only 0.5 mm precipitation from jointing to anthesis), which significantly reduced the maximum quantum efficiency of photosystem II, disrupting the chloroplast electron transport system and ultimately inhibiting photosynthesis (*Ehsanzadeh, Vaghar & Roushanzamir, 2021*). In particular, excessive N supply increased stomatal sensitivity to drought and ultimately increased stomatal limitation of the photosynthetic rate, to the detriment of crop photosynthesis (*Noor et al., 2023*). Despite the drought conditions, combined application of manure and topdressing N at 150 kg ha$^{-1}$ still resulted in the greatest increase in the photosynthetic rate of wheat. This is consistent with the notion that increased N supply *via* fertiliser application enhances flag leaf photosynthetic activity and carboxylation efficiency (*Effah et al., 2022*), promoting crop photosynthetic performance and dry matter production. Therefore, appropriate N supply increases photosynthetic resistance to drought stress by improving enzymatic or non-enzymatic antioxidant systems to mitigate drought stress-induced reductions in gas-exchange parameters (*Ru et al., 2024*); this further explains why photosynthesis remained highest in the M1N150 treatment under drought stress in the first year.

## Impact of fertiliser application on on grain yields, its components, NUA and NUE

The element that most affects crop growth and development is N, which plays vital roles in increasing crop yields and improving NUE (*Lyu et al., 2024*). The activity and reproduction of soil microorganisms promote the mineralisation of organic N in manure, producing large quantities of inorganic N for crop uptake and utilization (*Li et al., 2025*). In our 2-year field experiment, addition of manure increased grain yields, crop NUA, and NUE by 13.0%, 29.8%, and 27.5%, respectively (Fig. 4, Table 3). This aligns with findings that fertilisation substantially enhances yields and NUE by improving photosynthetic performance and facilitating N mobilisation in the stems and leaves of plants (*Sun et al., 2024*). In addition, N150 in combination with manure resulted in the highest grain yields and NUE, 2,123.4–2,838.8 kg ha$^{-1}$ and 20.7–39.2%, respectively (Fig. 4, Table 3), which may have been associated with sufficient N supply to improve leaf chlorophyll content, LAI values, photosynthetic efficiency, biomass, and NUA (*Rahman et al., 2000*;

*Samira, Parviz & Hassan, 2020*). The significant positive correlations between LAI values, aboveground biomass, crop NUA, and *Pn* values (2021–22) and grain yields and NUE in this study confirmed this (Fig. 6). However, yield and NUE decreased by 6.4% and 39.8%, respectively, when topdressing N rates with manure exceeded 150 kg ha$^{-1}$ (Fig. 4, Table 3). This is consistent with results obtained for dryland farmland in the Loess Plateau (*Liu et al., 2023*). Therefore, excessive N input can lead to reduced soil multifunctionality, which hinders wheat root development or increases permeability to the detriment of crop growth (*Li et al., 2024a*; *Li et al., 2024b*), bringing a risk of reductions in both yields and NUE.

Studies have found that manure and N application significantly improved grain yields by increasing SNs and GNSs (*Ranva et al., 2022*). In our study, SNs increased in both years and GNSs increased in the second year under the M1N150 treatment compared to N fertilisation alone (Table 3). The main reason may have been the synergistic effects of manure and N fertiliser application on the provision of sufficient nutrients and water, promoting root growth and ensuring that the wheat had enough tillers to increase the effective SN per unit area, boost growth and spike differentiation, thereby ultimately improving SNs and GNSs (*Dong et al., 2019*; *Zhang et al., 2024*). By contrast, application of manure and topdressing N fertiliser reduced the TGWs in both years and GNSs in the first year. The most immediate cause was a decrease in GNSs and TGWs due to drought (*Ru et al., 2024*), especially in the first year of drought stress. Water deficit hinders female reproductive development and ovule formation (*Mahrookashani et al., 2017*), while shortening the wheat grain filling period and disrupting the formation and accumulation of grain reserves (*Farooq, Hussain & Siddique, 2014*), all of which ultimately result in decreased GNSs and TGWs. In addition, the correlations between grain yields and SNs ($R^2 = 0.88$ and $0.79$, both seasons) and TGWs ($R^2 = -0.61$ and $-0.73$, both seasons) evident in Fig. 6 imply improved SNs to be the primary positive factor in increased grain yields, while TGWs are a negative factor that reduces yields in drought-prone areas.

## Impact of fertiliser application on SWS, ET$_c$ and WUE

In dryland agriculture in semi-arid areas, water is the most limiting factor in crop production. This study showed that the addition of manure increased ET$_c$ and WUE, but significantly decreased 0–100 cm SWS (Table 4, Fig. 5). A meta-analysis of the use of organic fertilisers in winter wheat production in northern China reached similar conclusions (*Wang et al., 2020*). In rainfed agricultural systems, water input and output are largely dependent on precipitation and ET$_c$, which determine the soil water balance and water availability (*Liu, Wu & Yang, 2022b*). *Li et al. (2022)* found that crop transpiration contributed the most to ET$_c$, and root uptake from the soil was the main pathway for water loss in crop transpiration. Therefore, differences in ET$_c$ and SWS may be due to fertilizer-induced effects on crops' root system (*Xing et al., 2025*). Studies have confirmed that organic fertilisers not only increase soil porosity and improve soil aggregates but also enhance soil microbial activity and promote nutrient decomposition and mineralization (*Karami et al., 2011*; *Liu et al., 2022a*), and that these changes benefit crop root growth. We found that ET$_c$ increased with the N application rate when N was added with manure, while SWS decreased; the highest WUE obtained was at N150 (Table 4, Fig. 5), in general agreement

with previous findings (*Li et al., 2023*). Studies have also shown that N is an important factor in root growth and distribution, and that the optimal N input significantly increases the root distribution area, activity, and water absorption capacity, thereby promoting crop growth (*Bai et al., 2021*). In the present study, $ET_c$ increased with the N application rate, while SWS decreased with increasing N application in combination with manure; the highest WUE was obtained at N150. These findings imply that combined application of manure and N fertiliser results in more water being consumed through transpiration as more biomass is synthesised, which leads to high $ET_c$ and WUE and low SWS (*Li et al., 2023*). Thus, application of manure in combination with an appropriate N rate is an effective regulatory measure to improve WUE.

## CONCLUSIONS

Our study underscores the vital roles played by manure and topdressing N management in wheat growth, grain yields, and efficient water and N use, all of which are pivotal for enhancing efficient agricultural resource use; this is advantageous even in drought-prone oasis dry cropping areas. A significant interaction involving WUE and NUE was observed between manure and N fertiliser. Addition of manure and topdressing with N fertiliser significantly promoted wheat growth and development and crop NUA, increasing $ET_c$ and leading to a decline in SWS. The greatest grain yields, WUE, and NUE were obtained with the M1N150 treatment. Compared to all treatments without added manure, the M1N150 treatment significantly increased SNs of dryland wheat, which effectively compensated for the negative effect of declining TGWs on yields. These results imply that the combination of manure at 30 t $ha^{-1}$ $yr^{-1}$ and topdressing N fertiliser at 150 kg $ha^{-1}$ was the best fertilisation strategy for improving the productivity of dryland winter wheat in Xinjiang, and constituted an efficient use of resources in this extremely arid environment.

### Funding

This study was funded by the National Natural Science Foundation of China (32060448), the Xinjiang Uygur Autonomous Region Graduate Student Research and Innovation Project (XJ2022G133), and the Key Discipline Project of Crop Science of Xinjiang Agricultural University (XNCDKY2021009). The funders had no role in study design, data collection and analysis, decision to publish, or preparation of the manuscript.

### Grant Disclosures

The following grant information was disclosed by the authors:
The National Natural Science Foundation of China: 32060448.
The Xinjiang Uygur Autonomous Region Graduate Student Research and Innovation Project: XJ2022G133.
The Key Discipline Project of Crop Science of Xinjiang Agricultural University: XNCDKY2021009.

## Competing Interests

The authors declare there are no competing interests.

## Author Contributions

- Yanfei Fang conceived and designed the experiments, performed the experiments, analyzed the data, prepared figures and/or tables, authored or reviewed drafts of the article, and approved the final draft.
- Jianghua Tang conceived and designed the experiments, analyzed the data, prepared figures and/or tables, authored or reviewed drafts of the article, and approved the final draft.
- Shanqing Zhang performed the experiments, prepared figures and/or tables, and approved the final draft.
- Na Zhang conceived and designed the experiments, authored or reviewed drafts of the article, and approved the final draft.
- Xiaoying Luo performed the experiments, authored or reviewed drafts of the article, and approved the final draft.
- Dongping Hu performed the experiments, prepared figures and/or tables, and approved the final draft.
- Wenxiu Xu conceived and designed the experiments, analyzed the data, prepared figures and/or tables, authored or reviewed drafts of the article, and approved the final draft.

## Data Availability

The raw measurements are available in the Supplementary File.

## Supplemental Information

Supplemental information for this article can be found online at http://dx.doi.org/10.7717/peerj.19543#supplemental-information.

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
