# Peer review of "Manure in combination with optimal topdressing with nitrogen fertiliser improved growth, grain yields and the efficiencies of water and nitrogen use in winter wheat in the Xinjiang Oasis drylands"

_PeerJ, doi:10.7717/peerj.19543_

## Round 0.1 · original submission · Major Revisions

Your manuscript still needs a major revision according to the reviewer's comments.

Reviewer 1 ·

Basic reporting

This study focus on the impact of manure and topdressing nitrogen fertiliser on the grain yield and water and nitrogen use efficiency of winter wheat in the drylands of the Xinjiang Oasis. The structure of the paper is well, but it has some shortcomings as mentioned in follows. So, I suggested the paper be accepted for publication after major improvement.
1.In the abstract part, the method, should be the experiment design, and the measured index, but not the contents of study.
2.In the introduction part, the more literature should be noted how manure and topdressing nitrogen fertiliser affect the productivity of dryland wheat, and explain the background of the study.
3.In the results part, the tables and figures should be add notes to increase its self-explanatory.
4. In the Discussion part, due to only two manure addition and three topdressing N rates, please strengthen the discussion of different fertilizer amounts, and explained why 30 t ha−1 of manure and 150 kg N ha−1 is proper.
5. There are many not clear expression and sentences, please polish the language for the whole paper.

Experimental design

no comment

Validity of the findings

no comment

Additional comments

no comment

·

Basic reporting

Review Report
A potentially good study has been conducted on effects of nitrogen application on winter wheat production. However, significant issues were detected in the manuscript that have to be addressed clearly. 

* The paper suffers from lack of correct analysis and soil water information and experimental Design.

* A wrong terminology has been used throughout the paper. The authors used water consumption term for crop evapotranspiration which is not correct. The water consumption is not acceptable for soil, water, and crop analysis.
* The statements in the introduction section do not have the appropriate connection. 
* The results of the other studies are not clear and very hard to be understood.
* The English language of the paper is strongly unacceptable. Even title language is not correct.

Experimental design

A field experiment was conducted from September 2021 to July 2023 in Sunjiagou Village, Mulei Kazakh Autonomous County, Changji Prefecture, Xinjiang Uygur Autonomous Region (43°83' N, 90°28'E). The study site is in a semi-arid rainfed agricultural region characterised by typical hilly-mountainous dryland. It is at an altitude of 1272 m, with an annual average air temperature of 6.6°C, annual sunshine duration of 3070 hours, frost-free period of 145 days, and average annual rainfall of 354 mm (1990–2023). The rainfall during the 2021–2022 and 2022–2023 growing seasons was 17% and 15% lower than the annual average, respectively. The soil type is dominated by chestnut-calcium (Chinese soil taxonomy). Table 1 details the physical and chemical properties of the 0–20 cm soil layer at the start of the experiment. Fig. 1 illustrates the monthly precipitation and average air temperature throughout the experiment. The precipitation at different stages of the experiment is detailed in Table 2.

Validity of the findings

Wenxiu Xu1
No.311 East Nongda Road, Bayi Street, Shayibak District, Urumqi, Xinjiang, 830052, China

Additional comments

Review Report
A potentially good study has been conducted on effects of nitrogen application on winter wheat production. However, significant issues were detected in the manuscript that have to be addressed clearly. 
1.The manuscript lacks a clear statement of specific research objectives, making it challenging to discern the research questions or hypotheses.
2.The experimental design and methodology lack detailed descriptions, hindering the assessment of study reliability and reproducibility.
3.The results and discussion sections lack in-depth analysis and interpretation, and expanding on the findings could provide a more comprehensive understanding of underlying mechanisms and implications
4.Consider measuring and reporting additional physiological and agronomic parameters, such as topdressing nitrogen fertiliser on the grain yield and nutrient uptake dynamics, and water-use efficiency, to enhance understanding of observed responses.
5.Conduct field trials across multiple seasons and locations to assess result consistency and generalizability under varying environmental conditions.
6.Explore the interactive effects of nitrogen management and other agronomic practices to optimize winter wheat production.
7.Perform economic and environmental analyses to evaluate the cost-effectiveness and sustainability of proposed nitrogen management strategies.
8. Strengthen the literature review and references to provide a more comprehensive understanding of the research area and highlight the study's contribution to existing literature.
* The paper suffers from lack of correct analysis and soil water information and experimental Design.

* A wrong terminology has been used throughout the paper. The authors used water consumption term for crop evapotranspiration which is not correct. The water consumption is not acceptable for soil, water, and crop analysis.
* The statements in the introduction section do not have the appropriate connection. 
* The results of the other studies are not clear and very hard to be understood.
* The English language of the paper is strongly unacceptable. Even title language is not correct.

Reviewer 3 ·

Basic reporting

The article is well-written and easily understood. However, the illustrations (Figures and tables) could be improved.

Experimental design

The experimental design is appropriate, but the factors and their levels are not suitable for the research's aim and title. For example, it must be explained why one dose of manure (30 t/ha) was used, as the research's aim is related to manure application.
The results of previous studies must also support it. The optimum levels of manure can not be determined by using one application dose. This study argues that the topdress application of nitrogen doses changes in the presence of manure fertilization.

Validity of the findings

The result should be improved and reevaluated.

Reviewer 4 ·

Basic reporting

1. The paper provides valuabe insights into sustainable farming in water-limited environments. However, the rigor of writing and data presentation did not allow m to accept this in its current shape . With extensive editing and also more robots presentation of the data may be need for re-evaluation
2. The treatments structure in the abstract should be provided, also the background in the abstract section is not clear, needs to be focus manure/N in semiarid area
3. Most of the abbreviation given in the abstract section (SWSJS-MS) are not clear, also no need to provided 2022-23 repeatedly.
4. The introduction provides a reasonable context but needs better emphasis on the novelty of the study. Highlighting the specific research gap in fertilizer application strategies for the Xinjiang Oasis is crucial. I did not find the research gap, or any hypothesis. There seem repetition in introduction, so many time yield and productivity effects have been given. It is not clear, that whether chemical or organic N improve soil fertility as mentioned in line 62. The objective given should be refined and based on factors and studied parameters.
5. Minor grammatical issues and typos exist that need addressing, especially in the discussion and results sections.
6. Neither the figure nor tables are clear, most of them is given interactive data, but the description is given on sole manure and N like we see in abstract (see line 25, 29, 33 etc). I donot think so that we need only significance in the two tables like tables like table 4 and 5, this should be either deleted or append to the specific table. Ensure datasets i.e. raw data are adequately described and accessible for replication.

Experimental design

1. The research question is relevant and addresses a significant gap in sustainable agriculture for arid regions.
2. Methods are detailed and reproducible. Specifics on manure composition and nitrogen topdressing timings are commendable.
3. However, the justification for manure rates and nitrogen splits could be linked more directly to previous findings.
L111, manure fertiliser added------------ repetition
L115, Both the cattle manure and 150 kg P2O5 (44% P) were applied as a basal fertiliser before=---------------- what do you mean by basal fertilizer mean
L147, the soil bulk density following Finn et al. (2015). Soil water storage (SWS) was calculated as follows (Luo et al., 2021):----------------- I donot think so, that these are original references who has develop these procedure, please provide the original references
L170, The root system was cut off and the plants were processed further.-------------- what do you mean by further process
L175, Nye.s colorimetric method--------------- what do you mean by Nye’s is this correct
L187, N is the amount of nitrogen fertiliser applied------------ which N you mean 150 from inorganic or total also from manure, how much was that

Validity of the findings

1. It is not clear, that whether the authors have reported only the interactive values or both main factors as well, the figure/tables contain only the interactive values, but the text also have main factors description in addition to the interactive. And also had reproduced each and every thing that is available in the figure and tables. It is advisable to report only the key and main finding directly linked to your objectives and hypothesis.
2. In discussion section, I did not see any specific mechanism, that why the addition of manure/N had positive effects, the current synthesis should be linked to the available literature and thus implication for the future should be made based on this data set.
3. Ensure that the discussion critically evaluates the broader implications of the findings, especially concerning global fertilizer use and water conservation strategies.

L197, significant (P < 0.05) or highly significant (P < 0.01) effects on SWS---------- use only one p values at least in a single statement
L198, extremely significant---------------- there is not extreme it will be p value
L199, and a significant effect at the maturity stage----------- confusing, what do you mean the rest of the interaction are non-significant

Additional comments

The manuscript is weak both writing-wise as well as data presentation-wise; both writing and data presentation should be improved

---

## Round 0.2 · accepted · Accept

Your changes are sufficient for your manuscript to be accepted. Congratulations

Reviewer 3 ·

Basic reporting

After corrections, it is suitable for publication.

Experimental design

After corrections, it is suitable for publication.

Validity of the findings

After corrections, it is suitable for publication.

Reviewer 4 ·

Basic reporting

no comments

Experimental design

no comments

Validity of the findings

No comments

Additional comments

The author have incorporated most of my comments, and I feel no hesitation to accept the manuscript in its current shape